# Genetic variation in the social environment affects behavioral phenotypes of oxytocin receptor mutants in zebrafish

Diogo Ribeiro[1], Ana Rita Nunes[1], Magda Teles[1], Savani Anbalagan[2,3], Janna Blechman[2], Gil Levkowitz[2], Rui F Oliveira[1,4,5]*

[1]Instituto Gulbenkian de Ciência, Oeiras, Portugal; [2]Weizmann Institute of Science, Rehovot, Israel; [3]ReMedy-International Research Agenda Unit, Centre of New Technologies, University of Warsaw, Warsaw, Poland; [4]ISPA – Instituto Universitário, Lisboa, Portugal; [5]Champalimaud Research, Lisboa, Portugal

**Abstract** Oxytocin-like peptides have been implicated in the regulation of a wide range of social behaviors across taxa. On the other hand, the social environment, which is composed of conspecifics that may vary in their genotypes, also influences social behavior, creating the possibility for indirect genetic effects. Here, we used a zebrafish oxytocin receptor knockout line to investigate how the genotypic composition of the social environment ($G_s$) interacts with the oxytocin genotype of the focal individual ($G_i$) in the regulation of its social behavior. For this purpose, we have raised wild-type or knock-out zebrafish in either wild-type or knock-out shoals and tested different components of social behavior in adults. $G_i$x$G_s$ effects were detected in some behaviors, highlighting the need to control for $G_i$x$G_s$ effects when interpreting results of experiments using genetically modified animals, since the genotypic composition of the social environment can either rescue or promote phenotypes associated with specific genes.

**\*For correspondence:**
ruiol@ispa.pt

**Competing interests:** The authors declare that no competing interests exist.

## Introduction

Social genetic effects (aka indirect genetic effects) occur when the phenotype of an organism is influenced by the genotypes of conspecifics. Previous work has highlighted the major potential evolutionary consequences of social genetic effects (*Moore et al., 1997*; *Wolf et al., 1998*), with evidence for such effects to be present both in interactions between related (e.g. mothers and offspring *Champagne and Meaney, 2006*; *Wilson et al., 2004*) and unrelated individuals (e.g. sexual displays (*Petfield et al., 2005*), aggression *Wilson et al., 2011*; *Sartori and Mantovani, 2013*; *Santostefano et al., 2017*). More recently, the importance of social genetic effects for health and disease has also been recognized (*Baud et al., 2017*), which may explain the pervasiveness of the social environment as a mortality risk in humans (*Holt-Lunstad et al., 2010*; *Holt-Lunstad et al., 2015*). Interestingly, the potential consequences of social genetic effects for the interpretation of research results using genetically modified organisms (GMO) has been greatly neglected. GMOs have been widely used in behavioral neuroscience to investigate the causal role of candidate genes and behavioral phenotypes. Typically Knock-in and Knock-out transgenics and mutants have been used to causally link the gain or loss of behavioral function to a specific gene (*Huang and Zeng, 2013*). In recent years, the development of genome editing techniques, such as CRISPR-Cas9-and TALEN-induced mutations, have increased the interest in this approach and opened the door to studying the genetic basis of behavior in non-model organisms (*Hsu et al., 2014*).

However, most studies using GMO in behavioral neuroscience have ignored the potential contribution of the genotypic composition of the social environment to the behavioral phenotype studied. This is because it has been assumed that if the genetic background of these mutants is identical and their environment has been kept constant, any phenotypic differences must come from the genetic manipulation. However, when GMOs are incrossed or visually screened at a very young age (e.g. using reporter genes, GFP) and thereafter raised and housed together until used in experiments, changes in their behavior might be affected by the divergent genotypic composition of social environments experienced by these mutants. In other words, modified behavior might be a result of growing with their peer mutants, rather than the canonical social environment provided by wild-type conspecifics. Such problem is particularly relevant when studying social behavior. Thus, given the rising interest in the study of social behavior in model organisms from worms to higher vertebrates, an assessment of the potential effect of the interaction between the genotype of the individual ($G_i$) and the genotypic composition of its social environment ($G_s$), on the behavioral phenotype of interest in GMOs used in social neuroscience is crucial.

Despite the wide variety of species-specific social behaviors, a wealth of evidence has implicated the paralog nonapeptides vasopressin (VP) and oxytocin (OXT) and their receptors in the regulation of different aspects of social behavior across vertebrates (*Donaldson and Young, 2008*; *Goodson and Thompson, 2010*), suggesting a genetic toolkit role (sensu evo-devo, i.e. ancient genes highly conserved among taxa that control the same biological process) for these nonapeptides in social behavior. Nonapeptides are an ancestral neuropeptide family found both in vertebrates and invertebrates, that derived from a VP-like peptide, and that evolved along two parallel clades of VP- and OXT-like peptides from the duplication of the VP gene in early jawed fish (ca. 500 Mya). Both peptides have been implicated in the regulation of behavior and physiology across different taxa, with VP being more involved in aggression and agonistic behaviors and OXT-like peptides consistently acting in affiliative behaviors and species-specific social behaviors across diverse taxa (i.e. sexual behavior, social interactions) (*Stoop, 2012*; *Goodson, 2013*). Despite this wealth of evidence on the direct genetic effects of OXT on social behavior, social genetic effects (i.e. $G_i$x$G_s$ effects) of OXT genotypes have never been studied.

In this study, we aimed to provide a proof of principle for $G_i$x$G_s$ effects in behavioral phenotypes observed in GMO by assessing the occurrence of such effects in a knockout line for the OXT receptor in zebrafish, a commonly used model species in behavioral neuroscience (*Orger and de Polavieja, 2017*), which forms social groups (aka shoals, *Miller and Gerlai, 2007*; *Miller and Gerlai, 2012*) and expresses a rich repertoire of social behavior (*Zebrafish Neuroscience Research Consortium et al., 2013*; *Nunes et al., 2017*). For this purpose, we studied the $G_i$x$G_s$ interaction in the effects of the OXT gene (*oxtr*) in different aspects of social behavior, by raising individual zebrafish of the WT (*oxtr*^(+/+)) or knock-out genotype (*oxtr*^(-/-)) in different social environments (i.e. *oxtr*^(+/+) shoal or *oxtr*^(-/-) shoal; *Figure 1A*). Since sociality encompasses motivational, cognitive and collective behavioral traits, we have selected a set of tests that aim to characterize these different aspects at a fundamental level: (*Moore et al., 1997*) the social preference and social habituation tests assess the motivation to approach conspecifics, and how it varies with the repeated access to conspecifics; (*Wolf et al., 1998*) the social recognition test, which provides an insight into the ability of zebrafish to discriminate between conspecifics based on one-trial learning; and (*Champagne and Meaney, 2006*) tests of shoaling behavior that assess how well the focal individual is able to integrate itself into an unfamiliar shoal and what influence it has on the behavior of the other shoal members.

## Results and discussion

Adult zebrafish, like many other social animals, express a tendency to approach and interact with conspecifics (social preference, *Figure 1B*; *Engeszer et al., 2004*). Here, we show that there was no significant effect of either genotype or $G_i$x$G_s$ interaction on social preference, but there was a marginally significant main effect of $G_s$ (*Table 1*; *Figure 1C*). When fish were presented for a second time to a shoal to measure social habituation (i.e. expected reduction in social preference), we found a $G_i$x$G_s$ interaction, where *oxtr*^(-/-) individuals raised in *oxtr*^(-/-) shoals express enhanced social habituation ($F_{1,44} = 5.642$, p=0.022; *Figure 1D*). Thus, social motivation in zebrafish seems to be influenced by the genotype of conspecifics rather than by the genotype of the individual. Hence, the increased social habituation in *oxtr*^(-/-) fish does not seem to be due to reduced social motivation, but rather to

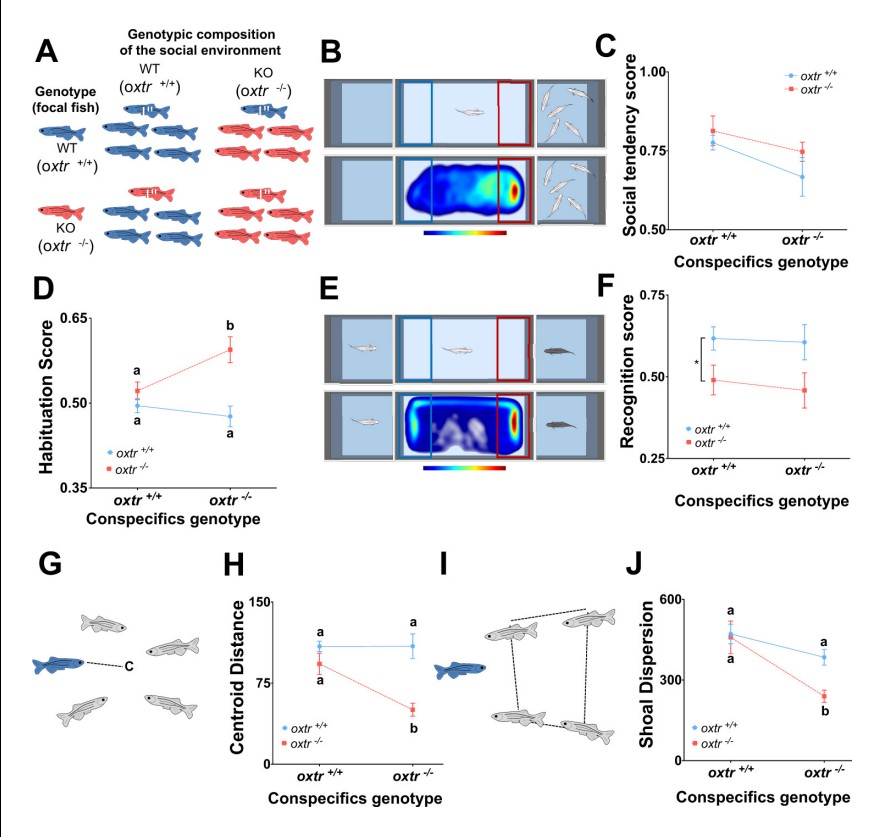

**Figure 1.** Genetic variation in the social environment affects zebrafish social behavior. The contribution of the individual genotype ($G_i$), the genotype of conspecifics in the social group ($G_s$) and the interaction between the two ($G_i \times G_s$) to the expression of behavioral phenotypes in zebrafish was assessed by raising oxytocin receptor mutant fish and wild types (focal fish marked with *) in shoals of either mutants or wild types (**A**). Social preference, measured by the time fish spend near a shoal vs. empty in a choice test (**B**, upper panel), showed a marginally significant effect of $G_s$ (**C**; Source data file *Figure 1—source data 1*). Social habituation, which consisted on a consecutive social preference test exhibited a $G_i \times G_s$ effect (**D**; Source data file *Figure 1—source data 2*). Social recognition, measured as the discrimination between a novel and a familiar conspecific (**E**, upper panel), shows a pure G effect (**F**; Source data file *Figure 1—source data 3*). Social integration, measured as distance to the centroid of the shoal (**G**), showed a $G_i \times G_s$ effect (**H**; Source data file *Figure 1—source data 4*). Social influence, measured by the cohesion of the remaining shoal members (**I**), also showed a marginally significant $G_i \times G_s$ effect (**J**; Source data file *Figure 1—source data 5*). Heatmaps show the spatial distribution of a representative $oxtr^{(+/+)}$ individual fish raised in a $oxtr^{(+/+)}$ group, during the entire trial, for both social preference (**B**, lower panel) and social recognition (**E**, lower panel). Data is presented as mean ± standard error of the mean (SEM). Sample sizes are nine for heterogeneous groups (i.e. focal individual with different genotype from the remaining individuals in the shoal; mutant focal in WT shoals and WT focal in mutant shoals) and 15 for homogeneous groups (i.e. focal individual with the same genotype of the remaining individuals in the shoal; mutant focal in mutant shoals and WT focal in WT shoals). Different letters indicate significant differences (p<0.05) between treatments as assessed by Tukey post-hoc tests following a two-way ANOVA (**D,H,J**; see *Table 1*). An asterisk indicates a $G_i$ main effect in **F**. The online version of this article includes the following source data for figure 1:

**Source data 1.** Effects of individual and conspecifics genotype on Social Preference.
**Source data 2.** Effects of individual and conspecifics genotype on social habituation.
**Source data 3.** Effects of individual and conspecifics genotype on social recognition.
**Source data 4.** Effects of individual and conspecifics genotype on social integration.
**Source data 5.** Effects of individual and conspecifics genotype on social influence.

**Table 1.** Effect of genotype of the focal individual ($G_i$), genotype of conspecifics present in its social environment ($G_s$) and the interaction between the two ($G_i$x$G_s$) on zebrafish social behavior was assessed using a two-way ANOVA.

~ indicates marginally significant, *p<0.05, **p<0.01, ***p<0.001. (Source data files *Figure 1—source datas 1–5*).

| Social preference | | | | | |
|---|---|---|---|---|---|
| | d.f. | Mean squares | F | Significance | Partial $\eta^2$ |
| $G_i$ | 1 | 0.023 | 1.731 | 0.195 | 0.038 |
| $G_s$ | 1 | 0.050 | 3.788 | 0.058~ | 0.079 |
| $G_i$ x $G_s$ | 1 | 0.001 | 0.049 | 0.825 | 0.001 |
| Error | 44 | 0.013 | | | |
| **Habituation** | | | | | |
| | d.f. | Mean squares | F | Significance | Partial $\eta^2$ |
| $G_i$ | 1 | 0.058 | 13.927 | 0.001 ** | 0.240 |
| $G_s$ | 1 | 0.008 | 1.936 | 0.171 | 0.042 |
| $G_i$ x $G_s$ | 1 | 0.024 | 5.642 | 0.022 * | 0.114 |
| Error | 44 | 0.004 | | | |
| **Social recognition** | | | | | |
| | d.f. | Mean squares | F | Significance | Partial $\eta^2$ |
| $G_i$ | 1 | 0.213 | 7.600 | 0.008 ** | 0.147 |
| $G_s$ | 1 | 0.005 | 0.189 | 0.666 | 0.004 |
| $G_i$ x $G_s$ | 1 | 0.001 | 0.041 | 0.841 | 0.001 |
| Error | 44 | 0.028 | | | |
| **Social group integration** | | | | | |
| | d.f. | Mean squares | F | Significance | Partial $\eta^2$ |
| $G_i$ | 1 | 39.486 | 24.370 | <0.001 *** | 0.356 |
| $G_s$ | 1 | 12.565 | 7.755 | 0.008 ** | 0.150 |
| $G_i$ x $G_s$ | 1 | 12.811 | 7.907 | 0.007 ** | 0.152 |
| Error | 44 | 1.620 | | | |
| **Social group dispersion** | | | | | |
| | d.f. | Mean squares | F | Significance | Partial $\eta^2$ |
| $G_i$ | 1 | 174.366 | 4.309 | 0.044 * | 0.089 |
| $G_s$ | 1 | 657.221 | 16.240 | <0.001 *** | 0.270 |
| $G_i$ x $G_s$ | 1 | 122.980 | 3.039 | 0.088 | 0.065 |
| Error | 44 | 40.469 | | | |

an heightened habituation to the stimuli, suggesting that the observed $G_i$x$G_s$ interaction effect is related to changes in single-stimulus learning mechanisms in mutant fish rather than to changes in social motivation.

When we tested social recognition, which is a form of social memory needed for individuality in social interactions (i.e. differential expression of social behavior depending on identity of interacting individual), that is known to be modulated by oxytocin both in mammals and zebrafish (*Ferguson et al., 2000*; *Ribeiro et al., 2020*), we observed that $oxtr^{(-/-)}$ individuals exhibit a deficit in acquisition and retention of social recognition irrespective of the social environment ($oxtr^{(-/-)}$ or $oxtr^{(+/+)}$) in which they were raised ($F_{1,44} = 7.600$, p=0.008; *Figure 1F*). Thus, in contrast to social motivation, social memory seems to rely on the individual's genotype. This result is in accordance with a recent study from our lab (*Ribeiro et al., 2020*) that has shown a deficit in one-trial recognition memory of both conspecifics and objects in $oxt^{(-/-)}$ fish, suggesting that this deficit is not specific to the social domain but is rather a general domain cognitive deficit.

Given that social behavior of zebrafish mainly occurs in the context of shoaling we have also investigated two shoaling behavior parameters: social integration and social influence. Social integration assesses how well the focal individual integrates in the social group (aka shoal), and is measured by its average distance to the centroid of the shoal (*Figure 1G,H*). A $G_i \times G_s$ interaction was found for social integration, where $oxtr^{(-/-)}$ individuals raised in $oxtr^{(-/-)}$ shoals exhibit a significantly lower social integration than $oxtr^{(-/-)}$ individuals raised in $oxtr^{(+/+)}$ shoals; in contrast, $oxtr^{(+/+)}$ individuals exhibit high levels of social integration irrespective of the shoal type in which they were raised (*Table 1*; *Figure 1H*). Social influence assesses how the focal individual affects the shoaling behavior of the remaining shoal members, by measuring the shoal dispersion as defined by the perimeter of the other shoal members (*Figure 1I,J*). The presence of a single WT ($oxtr^{(+/+)}$) individual in a $oxtr^{(-/-)}$ shoal was enough to increase its dispersion, whereas the presence of a single $oxtr^{(-/-)}$ individual in a $oxtr^{(+/+)}$ shoal did not affect its dispersion (*Table 1*; *Figure 1J*). In summary, we show that distinct components of social behavior are differentially affected by the genetic composition of the social environment versus the *oxtr* genotype of the focal individual. Social preference shows a marginally significant influence of the genotype of conspecifics. Social recognition exhibited a pure effect of the individual genotype. And clear $G_i \times G_s$ interactions were observed in the cases of social habituation and social integration. Social influence had a major contribution of the social environment, which is also the case, to a lesser extent, with social preference. Thus, we demonstrated that genetic variation in the social environment interacts with individual genotype during the developmental acquisition of social behavior. In other words, variation in the genotypes present in the social environment can revert particular phenotypes associated with specific genes. These results are in line with reported interactions between other aspects of the social environment and oxytocin receptor genotype in the determination of social behavior phenotypes in human populations (*Thompson et al., 2011*; *Wade et al., 2015*; *McQuaid et al., 2013*). Our results suggest that more caution is needed in the interpretation of studies using transgenic or mutant individuals that are raised in cohorts of the same genotype, and that some phenotypes observed in transgenic or mutant lines may in fact result from $G_i \times G_s$ interactions.

# Materials and methods

## Key resources table

| Reagent type (species) or resource | Designation | Source or reference | Identifiers | Additional information |
|---|---|---|---|---|
| Genetic reagent, TL (*Danio rerio*) | *oxtr* mutant line | *Nunes et al., 2020* | ZDB-ALT-190830–1 | |
| Commercial assay or kit | NucleoSpin Tissue | MACHEREY-NAGEL | # 740952.50 | For oxtr mutant genotyping |
| Sequence-based reagent | sense 5′-TGCGCGAGGAAAACTAGTT-3′ | Sigma | | For oxtr mutant genotyping |
| Sequence-based reagent | antisense 5′-AGCAGACACTCAGAATGGTCA-3′ | Sigma | | For oxtr mutant genotyping |
| Software, , algorithm | SPSS 25.0 | SPSS | RRID:SCR_002865 | |
| Software, , algorithm | Imagej (Fiji) | *Schindelin et al., 2012* | RRID:SCR_003070 | |
| Software, , algorithm | Ethovision XT 11.5 | Noldus Technology | www.noldus.com/ethovision | |
| Software, , algorithm | GraphPad Prism version 6.0 c | GraphPad software, San Diego, California, USA | www.graphpad.com | |
| Other | B and W mini surveillance camera | Henelec 300B | | Acquisition rate of 30 fps |
| Other | Webcameras | Logitech HD C525 | | Acquisition rate of 30 fps |

## Zebrafish lines and maintenance

Zebrafish were raised and bred according to standard protocols and all experimental procedures were approved by the host institution, Instituto Gulbenkian de Ciência, and by the National Veterinary Authority (DGAV, Portugal; permit number 0421/000/000/2013). OXTR mutant zebrafish line (ZFIN ID: ZDB-ALT-190830–1) was generated and provided by Dr. Gil Levkowitz (Weizmann Institute of Science) using a TALEN-based genome editing system. The characterization of this line has been described in *Nunes et al., 2020*.

All the experimental groups were formed at 4 days post-fertilization, based on the genotype of the progenitors, before they imprint for olfactory and visual kin recognition (*Gerlach et al., 2008*; *Hinz et al., 2013*). To evaluate genotype-environment effects, fish were raised in groups according to the experimental design in *Figure 1A* and both female and males tested in adulthood (3 months old). Sample sizes varied between nine for heterogeneous groups (i.e. focal individual with different genotype from the remaining individuals in the shoal) and 15 for homogeneous groups (i.e. focal individual with the same genotype of the remaining individuals in the shoal). The smaller sample size of heterogeneous groups is due to the need of genotyping all individuals in these groups to single out the focal individual.

## Genotyping

At 3 months old, 1-week before the behavioral screenings, genomic DNA was extracted from adult fin clips using the HotSHOT protocol (*Meeker et al., 2007*). All group members were fin clipped at different fin locations, to allow their identification while being maintained together. The genomic region of interest was amplified by PCR and sequenced to identify the focal fish in each group. The following primers were used: sense 5′-TGCGCGAGGAAAACTAGTT-3′, antisense 5′-AGCAGACAC TCAGAATGGTCA-3′.

## Behavioral assays

### Video acquisition

Fish were in a tank placed on top of an infrared lightbox and video-recorded either from above (shoal preference and social recognition tests) or laterally (group behaviour tests). Video acquisition was done with software Pinnacle Studio 14 (Corel Corporation, Ottawa, Canada). Shoal preference, social habituation and social recognition analyses were performed with EthoVision video tracking system (Noldus Information Technologies, Wageningen, The Netherlands) and group behavior analyses were done with the open source FIJI image-processing package (*Schindelin et al., 2012*).

### Social preference and social habituation

The social preference test assesses the individual's sociability by observing the interactions between conspecifics (*Ribeiro et al., 2020*): a focal fish was placed in a central compartment (30 × 15×10 cm) of a three-compartment tank, separated by transparent and sealed partitions. A shoal of unfamiliar fish was placed in one of the lateral compartments (15 × 10×10 cm), while the other contained only water. To avoid any side bias, the stimuli were balanced across trials. After an acclimatization period (10 min), the focal fish was released from a start box and allowed to explore the tank, while its behavior was video-recorded for 10 min. The time spent by the focal fish near (less than two body lengths) each compartment was quantified and used to calculate the social preference score ($SP$ = Time near shoal/ [Time near shoal + Time near empty]). A score above 0.5 indicates a preference for the shoal.

The social preference test was performed twice, with 24 hr in between, and social preference scores of both tests were used to calculate the habituation index (*Hab. Score* = 1- [$SP_{Trial2}$]/[$SP_{Trial1}$ + $SP_{Trial2}$]). A score above 0.5 represents a decrease in preference to associate with conspecifics.

## Social recognition

The social recognition assay to evaluate short-term (i.e. 10 min retention) social memory was adapted from the procedure already developed in our lab for long-term (i.e. 24 hr retention) social memory in zebrafish (*Gerlach et al., 2008*), and has already been used successfully in previous studies (*Ribeiro et al., 2020*; *Madeira and Oliveira, 2017*). A focal fish was placed for 10 min in the central compartment of a three-compartment tank, separated by transparent and sealed partitions, to

acclimatize. The focal fish was allowed to interact visually across partitions with two novel (unfamiliar) conspecifics for 10 min. After, both stimuli were removed, one was placed in the same compartment (familiar conspecific stimulus), while a novel conspecific was placed in the other compartment (novel conspecific stimulus). In a second 10 min interaction, the time spent by the focal fish near each compartment (termed novel cue or familiar cue) was quantified and used to measure the preference for the novel (*Recognition Score* = Time near Novel/[Time near Novel + Time near Familiar]). A recognition score of 0.5 indicates no preference between novel or familiar conspecifics.

### Shoaling behavior

Shoaling behavior is a common behavior present in fish models and allows to determine complex interactions between individuals. Both focal fish and social partners were recorded in the home tanks (3.5L tank). Focal fish were tagged with fin clips for easy identification. The behaviors were video-recorded from side view for 10 min. Two components of shoaling behavior were analyzed manually in time bins of 8 s, using FIJI software (*Schindelin et al., 2012*): (*Moore et al., 1997*) focal fish distance to the group centroid (social integration); and (*Wolf et al., 1998*) the dispersion of the remaining shoal members as measured by their perimeter (social influence).

### Data analysis

Data were analysed using SPSS 25.0. All data sets were tested for departures from normality with Shapiro-Wilks test. Two factor univariate ANOVA were used for comparing multiple groups. All data sets were corrected for multiple comparisons. Tukey's Test comparisons were used as post-hocs. Given that ANOVA is known to be underpowered for detecting significance of genotype x environment interaction (*Wahlsten, 1990*) we have decided to proceed with post-hoc tests for multiple comparisons among treatments even when $G_i$x$G_s$ interaction were only marginally significant (p<0.10). Graphs were performed with GraphPad software.

### Ethical approval

All experiments were performed in accordance with the relevant guidelines and regulations for the care and use of animals in research and approved by the competent Portuguese authority (Direcção Geral de Alimentação e Veterinária, permit 0421/000/000/2017).

## Acknowledgements

We thank IGC Fish Facility for assistance on fish maintenance and Peter McGregor for comments and discussion on an earlier version of the manuscript. The authors declare no conflicts of interest related to this work. This work was funded by a Fundação para a Ciência e a Tecnologia (FCT) research grant (PTDC/BIA-ANM/0810/2014) and a FEDER grant (Lisboa-01–0145-FEDER-030627) awarded to RFO. ARN and MST were supported by Post-doc fellowships from Fundação para a Ciência e a Tecnologia (FCT, ARN: SFRH/BPD/93317/2013). Congento LISBOA-01–0145-FEDER-022170, co-financed by FCT (Portugal) and Lisboa2020, under the PORTUGAL2020 agreement (European Regional Development Fund). The authors declare no competing interests.

## Additional information

### Funding

| Funder | Grant reference number | Author |
| --- | --- | --- |
| European Commission | Lisboa-01-0145-FEDER-030627 | Rui F Oliveira |
| Fundação para a Ciência e a Tecnologia | PTDC/BIA-ANM/0810/2014 | Rui F Oliveira |
| Fundação para a Ciência e a Tecnologia | SFRH/BPD/93317/2013 | Ana Rita Nunes Magda Teles |
| European Commission | LISBOA-01-0145-FEDER-022170 | Rui F Oliveira |

The funders had no role in study design, data collection and interpretation, or the decision to submit the work for publication.

## Author contributions
Diogo Ribeiro, Data curation, Formal analysis, Investigation, Visualization, Methodology, Writing - original draft, Writing - review and editing; Ana Rita Nunes, Data curation, Formal analysis, Investigation, Visualization, Methodology, Writing - review and editing; Magda Teles, Resources, Data curation, Formal analysis, Methodology, Writing - review and editing; Savani Anbalagan, Resources, Methodology; Janna Blechman, Resources, Funding acquisition, Methodology, Writing - review and editing; Gil Levkowitz, Conceptualization, Resources, Supervision, Funding acquisition, Methodology, Writing - original draft, Project administration, Writing - review and editing; Rui F Oliveira, Conceptualization, Resources, Data curation, Formal analysis, Supervision, Investigation, Writing - original draft, Project administration, Writing - review and editing

## Author ORCIDs
Gil Levkowitz (ID) http://orcid.org/0000-0002-3896-1881
Rui F Oliveira (ID) https://orcid.org/0000-0003-1528-618X

## Ethics
Animal experimentation: All experiments were performed in accordance with the relevant guidelines and regulations for the care and use of animals in research and approved by the competent Portuguese authority (Direcção Geral de Alimentação e Veterinária, permit 0421/000/000/2017).

## Decision letter and Author response
Decision letter https://doi.org/10.7554/eLife.56973.sa1
Author response https://doi.org/10.7554/eLife.56973.sa2

## Additional files

### Supplementary files
• Transparent reporting form

### Data availability
Data used in this study is provided as supplemental material at this stage. Data available on Dryad at doi: https://doi.org/10.5061/dryad.xwdbrv1bq.

The following dataset was generated:

| Author(s) | Year | Dataset title | Dataset URL | Database and Identifier |
|---|---|---|---|---|
| Oliveira RF | 2020 | Genetic variation in the social environment affects behavioral phenotypes of oxytocin receptor mutants | https://doi.org/10.5061/dryad.xwdbrv1bq | Dryad Digital Repository, 10.5061/dryad.xwdbrv1bq |

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
