## [Decision Letter]

**Acceptance summary:**

This paper asks a fundamental question regarding the influence of others' genotypes on the behavioral expression of an individual's genotype. The design is perfectly aligned to the question from the choice of the animal used to the conditions, genotypes and behavioral assays. The results are intrinsically interesting and also provide a critical proof of principle that will enable a myriad of future follow up investigations.

**Decision letter after peer review:**

Thank you for submitting your article "Genetic variation in the social environment affects behavioral phenotypes of oxytocin receptor mutants in zebrafish" for consideration by *eLife*. Your article has been reviewed by Peggy Mason as Reviewing Editor and Catherine Dulac as the Senior Editor, a Reviewing Editor, and three reviewers. The following individuals involved in review of your submission have agreed to reveal their identity: Peggy Mason (Reviewer #1); Lauren A O'Connell (Reviewer #2); Robert Gerlai (Reviewer #3).

The reviewers have discussed the reviews with one another and the Reviewing Editor has drafted this decision to help you prepare a revised submission.

The reviewers were excited about this clean study showing that the phenotypic expression of a genotype depends on the genotype of the individuals in the surrounding group as well as an individual's own genotype. This clear and fundamental study yielded an important result that is of broad interest to scientists trying to link genetics to behavior. It will inform both the design and interpretation of future work.

Reviewer #1:

This paper asks a fundamental question regarding the influence of others' genotypes on the behavioral expression of an individual's genotype. The design is perfectly aligned to the question from the choice of the animal used to the conditions, genotypes and behavioral assays. The results are intrinsically interesting and also provide a critical proof of principle that will enable a myriad of future follow up investigations.

My major comments are analytical and rhetorical. The only experimental question I have is what are the N-s? Please explicitly state how many fish were studied in each condition.

I disagree with the terminology used. Eg in the Abstract: the social environment, which is composed of conspecifics genotypes, NO, clearly the social environment is far more than the genotypes of those around an individual. And the issue is not really genetics vs environment. It is Genetics-self/individual vs Genetics-shoal. The environment, even the "social" environment, comprises far more than a fish's genetics: its age, reproductive status, nutritional/stress state and so on. In the Introduction, the authors say that previous works "have ignored the potential contribution of the environment to the behavioral phenotype studied." --This is not true. There is a large literature on how various forms of stress, occurring neonatally or at other times in the life cycle, interact with genotype. This example highlights the need to change the "environment" rhetoric used. It is, as the authors say: the genotypic composition of the social environment (Abstract). [Here I would only add the conspecific social environment as there are other species in an individual's natural environment.]

P=0.058 -> This P value is used in line Results and Discussion section to say there was no significant effect of either genotype, environment or GxEs interaction on social preference. This appears to be an overstated adherence to p<0.05 as meaningful. Is there a difference?

(SP = Time near shoal/ [Time near shoal + Time near empty]) -> why not just use Time near shoal? What is left out here is the time a fish spends in the middle of the middle chamber. A fish that spends 1 minite and one that spends 9 minutes next to the shoal would both be seen as low social preference if they spent no time next to the empty tank but these two fish would appear to differ. Additionally, the heat plot in Figure 1B in particular suggests that the proximity criterion is imposed upon the data arbitrarily and does not reflect the distribution of the fish's location. I suspect that a better metric could be devised in short order. Finally, it was unclear to me whether a similar criterion of proximity was used for the other metrics but if so, the same comment goes for them as for social preference.

The results would be far more accessible if the metrics were described in the text. One of the lovely features of *eLife* is that length is not an issue. Take the space you need to tell your story and no more is the guide rather than an arbitrary number of words. So please explain the metrics in the Results section. Even the explanations that are present could be expanded to great effect. For example, social preference is tendency to be near other fish rather than empty water (other choices for the non-preferred tank could conceivably be objects, a group of fewer fish, a single fish and so on). And while the metrics of habituation and individual recognition are clear (from the methods section), a word about the meaning of these metrics would be helpful. In other words, give the reader a reason to think that the metrics are important rather than simply measurable and therefore useful.

Reviewer #2:

This Short Report describes an elegantly designed experiment to show that the genotype of the rearing environment is important for behavior phenotypes of knockout animals in adulthood. They make the important point that the genotype of the rearing environment can influence behavioral phenotypes of knockout animals and cautions the scientific community to think about the genetic environment in which their knockouts are being raised, which will become even more important as more and more labs use knockout technologies in their animal of choice. Overall, I think this simple study is an important contribution to the field and have no major concerns.

Reviewer #3:

This is a brilliantly written, concise, logical and easy to follow paper in which the authors describe significant findings demonstrating GxE interaction in social behavior in zebrafish. The experiments are well described, well conducted, the results are well documented, illustrated, and are significant and important. The interpretation of the results is correct, and in addition to specifically for zebrafish social behavior research they also send a crucial message to all readers who study genetic effects on behavior.

1) Introduction.

Exactly because of the generality of the message, it'd be nice if the authors could expand their introduction/discussion and cite studies on zebrafish social behavior (e.g. Noam Miller had a series of experiments), genotype x environment interaction (with mice as well as with other fish species), as well as on the interpretation of studies using genetic manipulation (e.g. knock out). This broader intro would place the current study in the right context and would also correct the somewhat haphazard way the authors picked some studies to cite.

2) Materials and methods section.

Habituation score is not described for the social recognition test.

Shoaling behavior was recorded from a side-view camera, yet Figure 1G and I are depicting the fish from top view. Also, most shoaling studies using live shoal monitoring indeed monitor shoal cohesion, shoaling behavior with overhead cameras because shoal dispersion is better quantified in this two-dimensional plane, as opposed to from the side. One reason for this is that the relative distance of the fish to the top or to the bottom is more of a measure of fear/anxiety and less so of shoaling/social behavioral responses in laboratory tanks and in natural habitats with relatively shallow waters. Why was the side-view chosen?

Data analysis was conducted using ANOVA followed by Tukey's multiple range post-hoc tests. These are standard and accepted tests, but I note that ANOVA is known to be underpowered for detecting significance of interaction between its main factors. In fact, Wahlsten, (1990) demonstrated this specifically in the context of genotype x environment interaction. I draw the authors' attention to this because in their stat table, for example, they report G x E as p = 0.088 for social group dispersion, but find (I assume by Tukey) a clear indication of G x E interaction. It would be important for the reader to know the logic of how the authors proceeded from ANOVA to Tukey and how these seemingly conflicting stat findings are interpreted. That is, please cite the Wahlsten, 1990 paper and state what I mentioned above.

3) interpretation.

It is possible that the mutants do not recognize shoal-mates as conspecifics or do not have the ability to perceive and/or process finer social signals, hence the reduced discrimination score. There may be several other possible explanations too. Such possibilities of interpretive ideas/working hypotheses should be offered as they would help think about future studies for the analysis of behavioral as well as neurobiological mechanisms underlying the mutation induced changes in particular and the oxytocin-system in general.

Reviewer #4

The authors present an interesting paper that supports evidence for indirect genetics effects on social behaviour. They exploit the use of a zebrafish mutant for oxytocin receptor to understand if the social environment can modify the phenotypical response of the mutant in a plethora of different social behaviour assays.

The results of this paper have potentially a wide impact on the way social behavioural paradims are planned and executed. Therefore, the results need to be strong and heavily validated. I feel some controls and analysis could be added to support the main point. Below are my concerns about the paper:

1) Figure 1B and E

- I could not find on the legend what the heat map represents. Is it from a single fish? What type of fish and for how long?

2) Results Figure 1C.

- The social preference of w/t and OtxR fish raised with OtxR mutants is reduced. This is an important finding and it is not discussed in the paper. It seems that the social environment in both fish reduces sociality.

- The authors do not specify in the text if some of the fish used as social cue have been raised together with the focal fish.

- The authors identify mutated or w/t fish by genotyping the fish prior the experiments. How much time do the fish have to recover from the genotyping before the experiment, and are the fish selected for testing left in single tanks during this period of time?

3) Results Figure 1D.

- W/t fish seem to maintain similar habituation score when raised with w/t or OtxR mutants. In contrast, OtxR mutants seems to pay less attention to the social cue when exposed for the second time. This could be due to the fact that OtxR fish have a reduced interest for the social cue, however, other parameters could also be responsible for this. It is possible that the OtxR fish show a reduction of exploration to the chamber that is not specific to the social cue? Have the authors tested this possibility?

- Are the tested fish exposed to the exact same group of social cue fish after 24hours?

- The authors should analyse how w/t vs OtxR social cue interact differently with the test fish. This could provide an explanation for the difference seen in the graph results.

4) Results Figure G-J

- The authors say in the methods that they use tags to identify the tested fish. When was this tag introduced? The authors need to describe in detail what these tags are.

- My major concern is that the presence of a tag on a fish could also be the cause of the different behavioural phenotype seen. This could be resolved by using one of the several available tracking systems that allow to identify individual fish with high accuracy (Yun-Xiang et al., 2018, Romero-Ferrero et al., 2019).

- The authors in the methods write that a camera from the side was used to record the experiments. If this is the case, the schematics in Figure 1G and I should show fish from the side.

- "Two components of shoaling behaviour were analysed in time bins of 8 seconds". What is the frequency of recording of the experiments? Did the authors saw a difference over time?

Results and Discussion section: "In other words, the social environment can revert phenotypes associated with specific genes". This sentence is too strong since only few of the behaviours described in the paper seem to be modified by the social environment.

---

## [Author Response]

The reviewers were excited about this clean study showing that the phenotypic expression of a genotype depends on the genotype of the individuals in the surrounding group as well as an individual's own genotype. This clear and fundamental study yielded an important result that is of broad interest to scientists trying to link genetics to behavior. It will inform both the design and interpretation of future work.Reviewer #1:This paper asks a fundamental question regarding the influence of others' genotypes on the behavioral expression of an individual's genotype. The design is perfectly aligned to the question from the choice of the animal used to the conditions, genotypes and behavioral assays. The results are intrinsically interesting and also provide a critical proof of principle that will enable a myriad of future follow up investigations.

We thank very much the comments of the reviewer.

My major comments are analytical and rhetorical. The only experimental question I have is what are the N-s? Please explicitly state how many fish were studied in each condition.

The sample sizes in the 4 experimental conditions were: (1) wild-type in a wild-type shoal, n=15; (2) OxtR^+/-^ in an OxtR^+/-^ shoal, n=15; (3) Wild-type in a OxtR^+/-^ shoal, n=9; (4) OxtR^+/-^ in a wild-type shoal, n=9. This information was available in the legend of Figure 1. To make it more visible, we have also included this information in the Materials and methods section of the revised manuscript.

I disagree with the terminology used. Eg in the Abstract: the social environment, which is composed of conspecifics genotypes, NO, clearly the social environment is far more than the genotypes of those around an individual. And the issue is not really genetics vs environment. It is Genetics-self/individual vs Genetics-shoal. The environment, even the "social" environment, comprises far more than a fish's genetics: its age, reproductive status, nutritional/stress state and so on.

We totally agree with this remark. We have now realized that in some passages of the text one could get the impression that the social environment could be reduced to the genotypes of conspecifics, which is not what we meant. To avoid misleading the readers and following this recommendation of the reviewer, we have replaced GxE_s_ by G_i_xG_s_ in order to emphasize the contributions of the focal vs. conspecifics genotypes to the observed phenotypes. Hence, we hope to make clear, in this passage and along the whole manuscript, that the conspecifics’ genotypes are the component of the social environment that we are addressing in this paper, but that there are other dimensions of the social environment.

In the Introduction, the authors say that previous works "have ignored the potential contribution of the environment to the behavioral phenotype studied." --This is not true. There is a large literature on how various forms of stress, occurring neonatally or at other times in the life cycle, interact with genotype. This example highlights the need to change the "environment" rhetoric used. It is, as the authors say: the genotypic composition of the social environment (Abstract). [Here I would only add the conspecific social environment as there are other species in an individual's natural environment.]

Text has been changed accordingly.

P=0.058 -> This P value is used in line Results and Discussion section to say there was no significant effect of either genotype, environment or GxEs interaction on social preference. This appears to be an overstated adherence to p<0.05 as meaningful. Is there a difference?

As stated by the referee, the genotype of conspecifics main effect is marginally significant (p=0.058 with a partial partial η^2^ = 0.079, i.e. moderate effect), while no significant main effect is observed for genotype (p=0.195, partial η^2^ = 0.038, low effect) or G_i_xG_s_ interaction (p=0.825, partial η^2^ = 0.001, low effect)(note: we have used as reference for the magnitude of effect sizes: small for partial η^2^ > 0.01; medium for partial η^2^ > 0.06; and large for partial η^2^ > 0.14; http://imaging.mrc-cbu.cam.ac.uk/statswiki/FAQ/effectSize). To support the discussion of this analysis, we have now included in the Table 1 the partial eta squared for each test performed. Text has been changed accordingly.

(SP = Time near shoal/ [Time near shoal + Time near empty]) -> why not just use Time near shoal? What is left out here is the time a fish spends in the middle of the middle chamber. A fish that spends 1 min and one that spends 9 min next to the shoal would both be seen as low social preference if they spent no time next to the empty tank but these two fish would appear to differ.

We understand the reviewer concerns. However, if one uses the time near the shoal without taking in consideration the time near the empty tank we are not ruling out other physical factors of the empty tank in the attraction expressed by the focal fish towards the shoal that is presented inside a side compartment of the tank. For this reason, preference scores, such as the one we have used, are commonly used in the field, and are needed to assess the relative attraction towards the shoal after removing any putative effects of attraction towards an empty tank/ tank wall (e.g. thigmotaxis).

Additionally, the heat plot in 1B in particular suggests that the proximity criterion is imposed upon the data arbitrarily and does not reflect the distribution of the fish's location. I suspect that a better metric could be devised in short order.

This is an artifact of using different smoothing parameters to build the heatmap from the videotracking (using Ethovision). To illustrate this point we show in Author response image 1, Author response image 2, Author response image 3, Author response image 4 the original video tracking and its heatmap representation using different smoothing parameters. We have inadvertently used a very high smoothing (80) which have extended the high-density area outside the RoI. However, with lower smoothing values (e.g. 25, 50) the peak density of the spatial distribution of the fish is clearly within the RoI. We have now used a smoothing of 25 in Figure 1B.

**Author response image 1. sa2fig1:** 

**Author response image 3. sa2fig3:** 

**Author response image 4. sa2fig4:** 

Finally, it was unclear to me whether a similar criterion of proximity was used for the other metrics but if so, the same comment goes for them as for social preference.

An arbitrarily defined RoI was also used in the social recognition test, and subsequently an index (subsection “Social recognition”; *Recognition Score* = Time near Novel/[Time near Novel + Time near Familiar]) was used. For the same reason detailed above one cannot simply use the percentage of time spent with one of the two individuals and this is the score commonly used to measure the discrimination between the two stimuli fish. Anyway, we agree with the reviewer that in future studies it will be important to develop metrics of social preference and social recognition that are independent of arbitrarily defined RoIs.

The results would be far more accessible if the metrics were described in the text. One of the lovely features of eLife is that length is not an issue. Take the space you need to tell your story and no more is the guide rather than an arbitrary number of words. So please explain the metrics in the Results section. Even the explanations that are present could be expanded to great effect. For example, social preference is tendency to be near other fish rather than empty water (other choices for the non-preferred tank could conceivably be objects, a group of fewer fish, a single fish and so on). And while the metrics of habituation and individual recognition are clear (from the methods section), a word about the meaning of these metrics would be helpful. In other words, give the reader a reason to think that the metrics are important rather than simply measurable and therefore useful.

We have detailed more the rational for metrics used at the end of the Introduction. We acknowledged the recommendation for the use of more text as needed, although we are trying to follow the *eLife* recommendation that Short Reports should not exceed 1,500 words in the main text (excluding the Materials and methods section, References, and Figure legends).

Reviewer #2:This Short Report describes an elegantly designed experiment to show that the genotype of the rearing environment is important for behavior phenotypes of knockout animals in adulthood. They make the important point that the genotype of the rearing environment can influence behavioral phenotypes of knockout animals and cautions the scientific community to think about the genetic environment in which their knockouts are being raised, which will become even more important as more and more labs use knockout technologies in their animal of choice. Overall, I think this simple study is an important contribution to the field and have no major concerns.

We thank very much the commentaries of the reviewer.

Reviewer #3:This is a brilliantly written, concise, logical and easy to follow paper in which the authors describe significant findings demonstrating GxE interaction in social behavior in zebrafish. The experiments are well described, well conducted, the results are well documented, illustrated, and are significant and important. The interpretation of the results is correct, and in addition to specifically for zebrafish social behavior research they also send a crucial message to all readers who study genetic effects on behavior.1) Introduction.Exactly because of the generality of the message, it'd be nice if the authors could expand their introduction/discussion and cite studies on zebrafish social behavior (e.g. Noam Miller had a series of experiments), genotype x environment interaction (with mice as well as with other fish species), as well as on the interpretation of studies using genetic manipulation (e.g. knock out). This broader intro would place the current study in the right context and would also correct the somewhat haphazard way the authors picked some studies to cite.

We acknowledge the reviewer suggestion. However, *eLife* has a recommendation that the main text of Short Reports should not exceed 1,500 words (which includes Introduction, Results and Discussion section). Moreover, we have focused the Introduction in social genetic effects, which is a particular case of genotype x environment interaction (in which the genotype of conspecifics present in the environment is the environmental component that is taken into consideration), and we would like to keep it focused instead of opening it to general GxE effects, for which there is an abundant literature. Nevertheless, in the revised text we cite some of the suggested literature on zebrafish social behavior, without increasing significantly or changing the tone of the Introduction.

2) Materials and methods section.Habituation score is not described for the social recognition test.

We have not measured habituation in the social recognition test since the construct of habituation in the scope of discrimination learning is not established in the literature. In fact, both habituation and recognition represent two different types of learning (the former is a single stimulus learning and the latter a two-stimuli learning). Thus, the social recognition test was performed only once and we reported the social recognition score which translates the ability of the fish to discriminate between a familiar vs. a novel fish, which is what is commonly reported in this type of study in other species, such as rodents.

Shoaling behavior was recorded from a side-view camera, yet Figure 1G and I are depicting the fish from top view. Also, most shoaling studies using live shoal monitoring indeed monitor shoal cohesion, shoaling behavior with overhead cameras because shoal dispersion is better quantified in this two-dimensional plane, as opposed to from the side. One reason for this is that the relative distance of the fish to the top or to the bottom is more of a measure of fear/anxiety and less so of shoaling/social behavioral responses in laboratory tanks and in natural habitats with relatively shallow waters. Why was the side-view chosen?

We agree with the reviewer, but in we recorded the fish in their home tanks not only to reduce fish manipulation and consequently, stress induced by the manipulation, but mainly for us to be able to identify the focal fish within a group (all fishes were fin clipped at different locations to allow the identification of the focal fish after genotyping). In this respect, it would be impossible to identify the focal fish if the recording was done from a top view. We have now changed the Figure 1G and I to reflect the recording from a side-view.

Data analysis was conducted using ANOVA followed by Tukey's multiple range post-hoc tests. These are standard and accepted tests, but I note that ANOVA is known to be underpowered for detecting significance of interaction between its main factors. In fact, Wahlsten, (1990) demonstrated this specifically in the context of genotype x environment interaction. I draw the authors' attention to this because in their stat table, for example, they report G x E as p = 0.088 for social group dispersion, but find (I assume by Tukey) a clear indication of G x E interaction. It would be important for the reader to know the logic of how the authors proceeded from ANOVA to Tukey and how these seemingly conflicting stat findings are interpreted. That is, please cite the Wahlsten, 1990 paper and state what I mentioned above.

Following this commentary we have revised the use of post-hoc Tukey tests in this study following the ANOVA main effects and interaction tests. Given the low power of ANOVA for detecting significance of interaction between its main factors mentioned by the reviewer, we have decided to proceed with post-hoc tests when the p-value for the interaction was marginally significant (p<0.10). Otherwise, we have not proceeded with the post-hoc analysis for the interaction. As a result we have changed Figure 1F where there was an effect of genotype but not of the interaction, and we kept the post-hoc analysis in Figure 1J, where there was a marginally significant G_i_xG_s_ interaction (p=0.088). Moreover, we have now also included the partial eta squared in the revised Table 1 and the marginally significant G_i_xG_s_ interaction reported shows a moderate partial η^2^ of 0.065.

3) interpretation.It is possible that the mutants do not recognize shoal-mates as conspecifics or do not have the ability to perceive and/or process finer social signals, hence the reduced discrimination score. There may be several other possible explanations too. Such possibilities of interpretive ideas/working hypotheses should be offered as they would help think about future studies for the analysis of behavioral as well as neurobiological mechanisms underlying the mutation induced changes in particular and the oxytocin-system in general.

In this paper we aim to draw attention to potential G_i_xG_s_ effects in GMO animals used in behavioral neuroscience, and we used the oxt-receptor (*oxtr*) mutant as a case study. Thus, in the writing of the paper we are more focused on establishing a proof of concept rather than providing detailed tentative hypothesis for the observed results. Nevertheless, regarding the genotypic effect observed in *oxtr*^-/-^ fish, it has already been reported in another recent study from our lab (Ribeiro et al., 2020), where we have shown that this deficit is also present for object recognition, hence suggesting a general recognition memory deficit rather than a specific deficit in the social domain. We have added this information to the revised text.

Reviewer #4The authors present an interesting paper that supports evidence for indirect genetics effects on social behaviour. They exploit the use of a zebrafish mutant for oxytocin receptor to understand if the social environment can modify the phenotypical response of the mutant in a plethora of different social behaviour assays.The results of this paper have potentially a wide impact on the way social behavioural paradims are planned and executed. Therefore, the results need to be strong and heavily validated.

We thank the reviewer for the commentaries.

I feel some controls and analysis could be added to support the main point. Below are my concerns about the paper:1) Figure 1B and E- I could not find on the legend what the heat map represents. Is it from a single fish? What type of fish and for how long?

The heat map is from a single representative WT focal fish raised in a wild-type social environment during a 10minute trial in both social preference (1B) and social recognition (1E). We have now included this information in the figure legend.

2) Results Figure 1C.- The social preference of w/t and OtxR fish raised with OtxR mutants is reduced. This is an important finding and it is not discussed in the paper. It seems that the social environment in both fish reduces sociality.

As mentioned above by referee 1, there is a marginally significant main environment effect on social preference (p=0.058). We have now highlighted this finding in the last paragraph of the Results section.

- The authors do not specify in the text if some of the fish used as social cue have been raised together with the focal fish.

Fish used as social cue were always unfamiliar fish which were not raised together with the focal fish. This is applicable for both Social Preference and first part of the social recognition test. We have now included this information in the subsection “Social preference and social habituation”.

- The authors identify mutated or w/t fish by genotyping the fish prior the experiments. How much time do the fish have to recover from the genotyping before the experiment, and are the fish selected for testing left in single tanks during this period of time?

Fish were genotyped by fin cliping one-week prior to the experiments, to allow them to recover before being tested. During this period of time, the fish selected for testing were always together with their group members, which was possible because all members of the group were fin clipped at specific fin locations. To make it clear, we have now included this information in the subsection “Genotyping”.

3) Results Figure 1D.- W/t fish seem to maintain similar habituation score when raised with w/t or OtxR mutants. In contrast, OtxR mutants seems to pay less attention to the social cue when exposed for the second time. This could be due to the fact that OtxR fish have a reduced interest for the social cue, however, other parameters could also be responsible for this. It is possible that the OtxR fish show a reduction of exploration to the chamber that is not specific to the social cue? Have the authors tested this possibility?

We have also computed stimuli exploratory scores (stimuli exploratory score = (T shoal + T empty) / T total). As can be observed in the graphs bellow, there were no differences in stimuli exploratory score between the different groups in the two tests performed for social preference (Test 1- genotype: p=0.206, partial η^2^ 0.036; environment: p=0.992, partial η^2^ 0.000; G_i_xG_s_ interaction: p=0.867, partial η^2^ 0.001; Test 2- genotype: p=0.802, partial η^2^ 0.001; environment: p=0.336, partial η^2^ 0.021; G_i_xG_s_ interaction: p=0.686, partial η^2^ 0.004).

**Author response image 5. sa2fig5:** 

Furthermore, as shown in the following graphs, there are no differences in mean speed of the different groups during the two tests of social preference performed (Test 1- genotype: p=0.794, partial η^2^ 0.002; environment: p=0.492, partial η^2^ 0.011; G_i_xG_s_ interaction: p=0.703, partial η^2^ 0.003; Test 2- genotype: p=0.812, partial η^2^ 0.001; environment: p=0.516, partial η^2^ 0.010; G_i_xG_s_ interaction: p=0.202, partial η^2^ 0.037).

**Author response image 6. sa2fig6:** 

- Are the tested fish exposed to the exact same group of social cue fish after 24hours?

After 24 hours, fish were exposed to a different shoal of conspecifics in the same behavioral setup.

- The authors should analyse how w/t vs OtxR social cue interact differently with the test fish. This could provide an explanation for the difference seen in the graph results.

Although we agree that this analysis could be very interesting, it is currently not possible because the shoal of conspecifics were not recorded during the trial, only the focal fish.

4) Results Figure G-J- The authors say in the methods that they use tags to identify the tested fish. When was this tag introduced? The authors need to describe in detail what these tags are.

We clearly explain in the revised text that fish were tagged through fin clipping at different fin locations in order to genotype and identify the focal fish, one-week prior to the behavioral experiment. The following fin clips were performed within each group:

Fish 1- Bottom caudal fin

Fish 2- Top caudal fin

Fish 3- Bottom and Top caudal fin

Fish 4- Bottom caudal and anal fin

Fish 5- Top caudal and anal fin

- My major concern is that the presence of a tag on a fish could also be the cause of the different behavioural phenotype seen. This could be resolved by using one of the several available tracking systems that allow to identify individual fish with high accuracy (Yun-Xiang et al., 2018, Romero-Ferrero et al., 2019).

We believe that the fin clipping was not the cause of the different behavioral phenotypes observed since: (1) all fish from the group were fin clipped at different fin locations to allow their identification; (2) the fin clip of the focal fish was not always in the same position; and (3) fish were allowed one-week recovery before being tested.

- The authors in the methods write that a camera from the side was used to record the experiments. If this is the case, the schematics in Figure 1G and I should show fish from the side.

Figure has been changed accordingly.

- "Two components of shoaling behaviour were analysed in time bins of 8 seconds". What is the frequency of recording of the experiments? Did the authors saw a difference over time?

The fish were recorded during the entire trial of 10 min, but the analyses were done every other 8 seconds. No differences were observed over time, as shown in Author response image 7 and Author response image 8 for focal fish distance to group centroid and shoal dispersion.

**Author response image 7. sa2fig7:** 

**Author response image 8. sa2fig8:** 

Results and Discussion section: "In other words, the social environment can revert phenotypes associated with specific genes". This sentence is too strong since only few of the behaviours described in the paper seem to be modified by the social environment.

The reviewer is absolutely right, we have changed the text to: “(…) the social environment can revert particular phenotypes associated with specific genes.”